ⓜ | **Open Peer Review** | Biotechnology | Research Article

# Genome-scale metabolic modeling of *Ruminiclostridium cellulolyticum*: a microbial cell factory for valorization of lignocellulosic biomass

Idun Burgos,[1] Ove Øyås,[2,3] Stéphanie Perret,[4] Henri-Pierre Fierobe,[4] Daniel Machado[1]

**ABSTRACT** The development of sustainable biotechnological processes requires a transition from the traditional fermentation of refined substrates toward the valorization of waste materials such as lignocellulosic biomass. Although these so-called recalcitrant substrates cannot be degraded by model industrial organisms, they can be degraded by microbial consortia through a process of anaerobic digestion, where different community members are able to break down polysaccharides of varied complexity. Among these microbes, *Ruminiclostridium cellulolyticum* stands out as a promising candidate for fermentation of lignocellulose due to its ability to degrade both cellulose and hemicellulose. In this work, we present an updated genome-scale metabolic model for *R. cellulolyticum* strain H10. The model was manually curated with experimental data, and the pathways for degradation of cellulose and hemicellulose (arabinoxylan and xyloglucan) were reconstructed and annotated with full detail. The model enables the simulation of the fermentation profile of lignocellulosic materials of various compositions, facilitating the use of this organism as a potential workhorse for sustainable biotechnology, and it provides a valuable template for the reconstruction and optimization of lignocellulose degradation pathways in related organisms.

**IMPORTANCE** In this work, we present a manually curated genome-scale metabolic model for *Ruminiclostridium cellulolyticum*, one of the few species known to fully degrade cellulose and hemicellulose. The model was extensively curated with experimental data obtained from the literature, covering approximately 25 years of research on this organism. We use this model to simulate the fermentation of mixed lignocellulosic polysaccharides and observe a good agreement with experimental data. This organism is therefore a promising microbial cell factory for sustainable transformation of lignocellulosic residues into valuable industrial products.

**KEYWORDS** metabolism, modeling, genome-scale models, flux balance analysis, cellulose, lignocellulose, fermentation, biotechnology

Anaerobic digestion, in which bacterial communities degrade and ferment biowaste, is a popular approach for biowaste valorization (1). It is based on a natural process that occurs in, for example, the rumen and the soil and can yield products like biogas and other valuable fermentation products. However, controlling the anaerobic digestion process is a challenge due to the variability in substrate composition, community structure, inhibitory substances, and metabolic bottlenecks (1). Due to the intricate nature of the community's metabolic capabilities and interactions among its members, controlling the carbon flow toward a desired end product remains a challenge (2, 3). This challenge is especially acute given the diversity of metabolic capabilities within the community, even though certain functional guilds are conserved (4). One way to

Address correspondence to Daniel Machado, daniel.machado@ntnu.no.

The authors declare no conflict of interest.

See the funding table on p. 14.

improve control is through microbial community modeling. This approach, based on genome-scale metabolic models (GEMs), has received much attention because of its ability to give mechanistic insights and predict metabolic interactions (5).

The hydrolysis of cellulose and hemicellulose, the polysaccharide components of lignocellulose, has been shown to be the rate-limiting step or metabolic bottleneck of anaerobic digestion (6). There are only a few characterized organisms that have the ability to degrade and use these polysaccharides as a carbon source, due to the recalcitrant structure and complexity of such compounds (7). One of these organisms is the gram-positive mesophilic bacterium *Ruminiclostridium cellulolyticum*, which has the inherent ability to reduce both hemicellulose and cellulose into their structural components (8). It produces a multi-enzyme complex called the cellulosome, typical of the Clostridia growing on crystalline cellulose, which efficiently catalyzes extracellular degradation of lignocellulose. *R. cellulolyticum* is an anaerobic heterotroph with a fermentation profile consisting mainly of acetate, ethanol, lactate, hydrogen, and $CO_2$ (9).

Although other cellulolytic Clostridia like the thermophile *Clostridium thermocellum* and *Clostridium cellulovorans* are also being studied and tested as potential cell factories (10), there are some clear advantages to the application of *R. cellulolyticum*. First, it is capable of metabolizing a broad spectrum of sugars and oligosaccharides derived from both the cellulose and hemicellulose fractions of lignocellulose (11), whereas other Clostridia show a more limited substrate spectrum (12). Second, most of the polysaccharide-degrading enzymes are associated with the cellulosome, a feature that contributes to high degradation efficiency (13). Finally, a genetic toolbox, with several vectors for inactivation and overexpression of genes, has recently been developed (14–16). All of these factors combined make *R. cellulolyticum* a promising microbial cell factory for the valorization of lignocellulose.

The first GEM for *R. cellulolyticum* strain H10, iFS431, was constructed by Salimi et al. (17), following the publication of its genome in 2009 (18). The model accounts for 603 metabolites, 621 reactions, and 431 genes. It was built largely based on the annotations available in the JGI database (19). One of the major limitations of the iFS431 model is that it does not incorporate the pathways for degradation of complex polysaccharides other than cellulose, and the pathway for cellulose degradation is simplified compared to the current knowledge of all the steps involved. This stems from the fact that these pathways have only recently been characterized and annotated in detail (11, 20–27). In the meantime, GEMs for other cellulolytic Clostridia have also been published. *C. thermocellum* is considered a model species for cellulose-degrading bacteria and has gained much interest for its ability to produce ethanol from cellulose. There are multiple genome-scale metabolic models for this organism (28–30) with sequential improvements between versions, including more detailed cellulose degradation pathways.

In this work, we present a genome-scale metabolic model for *R. cellulolyticum* H10, denoted as iIB727. After an automated genome-based reconstruction, the model was extensively curated based on experimental data obtained from the literature, including growth on multiple substrates, fermentation profiles, cofactor specificity, and mutant phenotypes. In addition, the pathways for degradation and uptake of cellulose, xyloglucan, and arabinoxylan were reconstructed and annotated at a high level of detail by incorporating recent findings. We demonstrate the applicability of the model in the simulation of batch fermentations using lignocellulosic materials with different compositions. The ability to predict the impact of genetic and environmental perturbations provides a suitable framework to transform *R. cellulolyticum* into a workhorse of industrial biotechnology.

## RESULTS

### Draft model reconstruction

An initial draft reconstruction of *R. cellulolyticum* H10 was created using CarveMe (31), based on the only available genome of this species in RefSeq (GCF_000022065.1, a high-quality complete assembly). Figure 1 shows a summary of the genes, reactions,

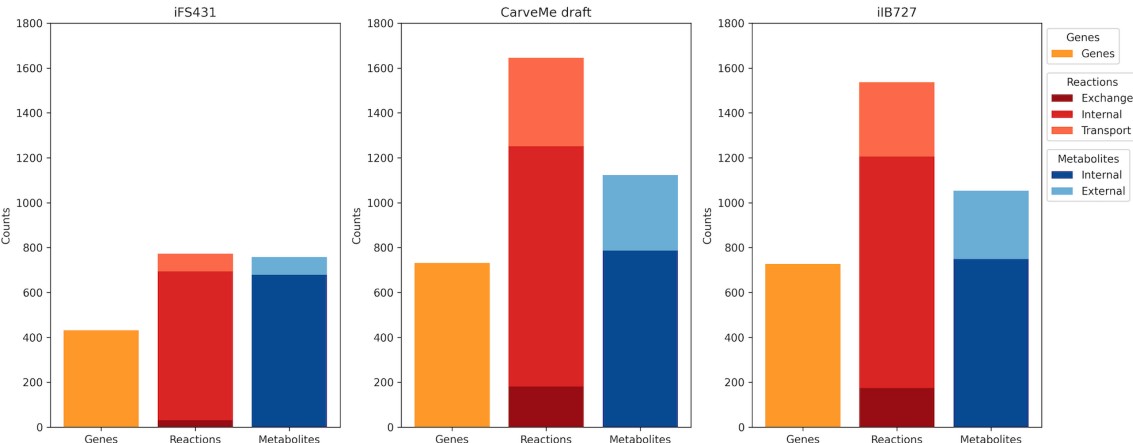

FIG 1  Summary of metabolic models. The total number of genes, reactions, and metabolites is described for the earlier iFS431 model, the initial draft produced with CarveMe, and the final version of the manually curated model, iIB727.

and metabolites present in the iIB727 model in comparison with the older iFS431 model. Interestingly, CarveMe automatically reconstructed the pathways for xylan (β-1,4 with glucuronic acid decorations), glucomannan, galactomannan, and glucan (β-1,3) degradation. These are mostly derived from the *Thermotoga maritima* model iLJ478 (32) present in the BiGG database.

We evaluated the consistency and annotation of all models using the MEMOTE test suite (33). The curated iIB727 model got a total MEMOTE score of 88%, which is a significant improvement compared to iFS431 (total score 36%), and the initial draft model (70%). Briefly, the final MEMOTE score of 88% is a weighted combination of multiple subscores, which represent different evaluation metrics, including one for consistency and others for annotation level. The subscore for consistency is composed of stoichiometric consistency (100.0%), mass balance (99.9%), charge balance (78.8%), and metabolite connectivity (100.0%).

As described in Materials and Methods, media on which *R. cellulolyticum* has been reported to grow (11, 22, 23, 26, 34–36) were used for gap-filling during the reconstruction process. These are mainly minimal media with variations of the carbon source, from glucose and other monosaccharides to cellulose and other polysaccharides. We used flux balance analysis (FBA) to predict the maximum growth rate on each medium (Fig. S1). We can see that neither iFS431 nor the CarveMe draft model is able to utilize all substrates due to the lack of the respective degradation pathways, whereas iIB727 can reproduce growth across all media.

## Manual reconstruction of cellulose and hemicellulose degradation

The most studied pathway for degradation of lignocellulosic components is that of cellulose metabolism. There are extensive studies of both extracellular and intracellular enzymes involved in this pathway (24, 25, 37, 38). Desvaux described that oligosaccharides of cellulose, cellodextrins, with a length of up to seven glucose units can be imported and degraded in the cytosol (9). In recent studies of the ABC transporter for cellodextrins, they reported activity on cellodextrins up to a length of five glucose units (25). We reconstructed the transport reactions according to their findings. After being imported into the cytosol by an ABC transporter, the cellodextrins are degraded sequentially by cellobiose/cellodextrin phosphorylases (CEPA). Recent studies have shown that the CEPAs are highly specific to the length of the cellodextrin (24). The pathway for cellodextrin uptake and degradation was manually reconstructed according to the information above and incorporated into the model (Fig. 2). All reactions were able to carry flux as long as the corresponding cellodextrin was provided.

In addition to cellulose, the metabolism of tamarind xyloglucan has also been extensively studied (11, 22, 40, 40, 41). This metabolism is more complex than that of cellulose due to the composition of this branched heteropolysaccharide. The backbone of xyloglucan is the same as that of cellulose, but the glucosyl units are highly decorated. The xyloglucanases that degrade the polysaccharide extracellularly typically degrade them into oligosaccharides with a glucose backbone length of four units (22). After transport through an ABC transporter, intracellular degradation of xyloglucan polysaccharides starts with a step-by-step removal of the decorations of galactose and xylose on the glycosyl residues of the backbone. The first step is the removal of all galactose decorations by β-galactosidase, followed by the removal of xylose, and then glucose, from the reducing end by α-xylosidase and β-glucosidase, respectively. Using this information, we reconstructed the pathway for xyloglucan degradation and combined it with the model in a similar manner to that of cellulose (Fig. 2). In the model, the letters G, Q, or L in the metabolite identifiers for oligosaccharides indicate the type of decoration on a glucosyl residue in the backbone. "G" denotes a glucosyl residue

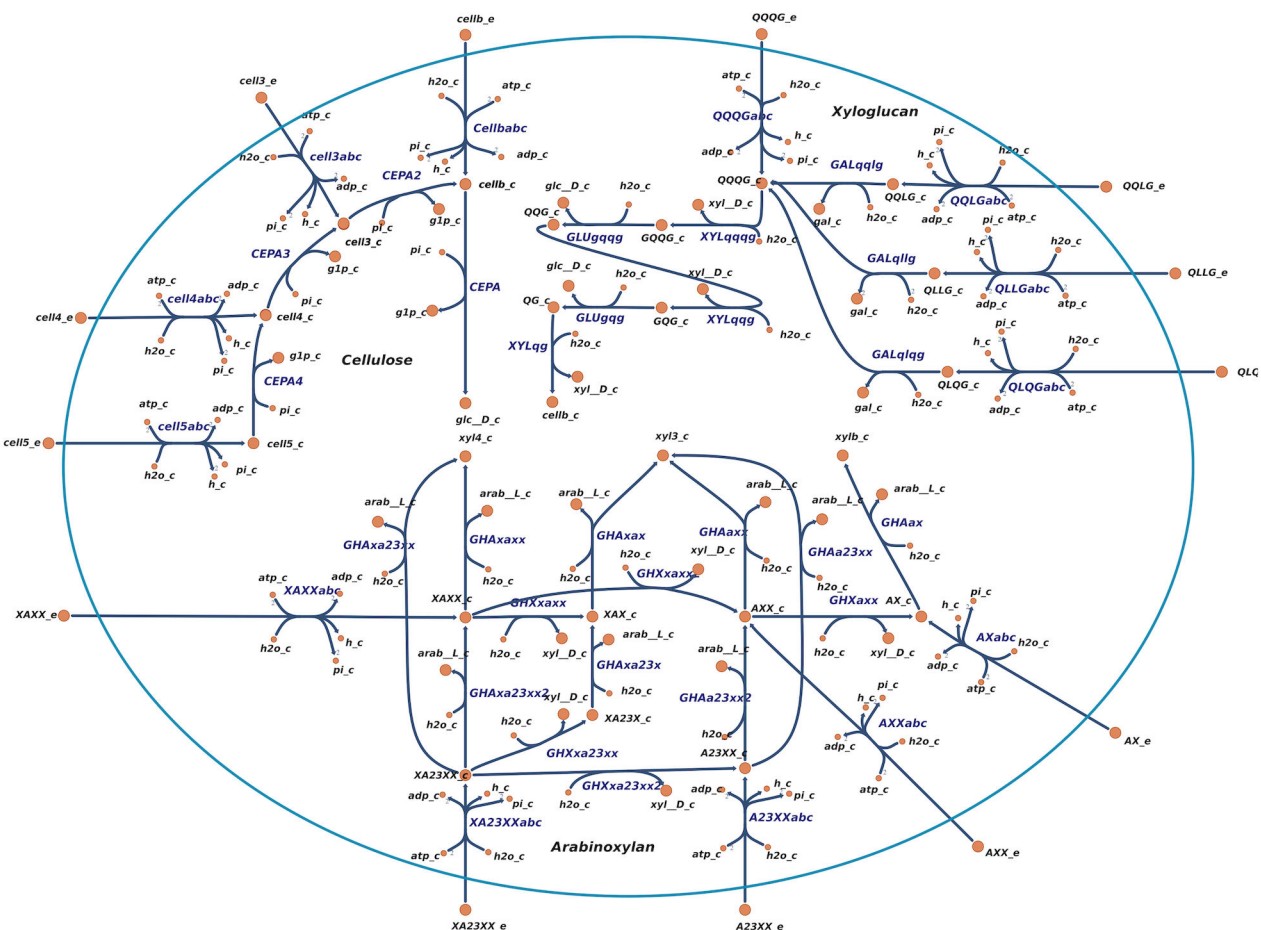

FIG 2 The cellulose degradation pathway (top left) includes degradation pathways for oligosaccharides of different sizes: cellb (cellobiose, two monomer units), cell3, cell4, and cell5 (with three, four, and five units, respectively). The arabinoxylan degradation pathway (bottom) starts with the transport of AX, AXX, A23XX, and XA23X oligosaccharides across the membrane by ABC transporters. Their degradation is illustrated with right-directed edges for xylanases (IDs starting with GHX), which act on the main chain of β-1,4-linked xylose units, and upward-directed edges for arabinoxylanases (IDs starting with GHA), which remove α-1,2- or α-1,3-linked arabinose decorations. XAX, XA23X, and AXX are intermediate products of intracellular degradation. The xyloglucan degradation pathway (top right) starts with the transport of QQQG, QQLG, QLLG, and QLGG oligosaccharides across the membrane by ABC transporters. The degradation involves a combination of β-glucosidases (GLU*), α-xylosidases (XYL*), and β-galactosidases (GAL*). The β-glucosidases target the main chain of β-1,4-glucose molecules, while the others target the decorations of xylose and galactose. QQG and QG are intermediate products of intracellular degradation. Metabolite abbreviations: glc__D (D-glucose), gal (D-galactose), xyl__D (D-xylose), xylb (xylobiose), xyl3 (xylotriose), xyl4 (xylotetraose), arab__L (L-arabinose), g1p (glucose-1-phosphate), and pi (phosphate). This map was generated using Escher (39).

without any decoration, "Q" denotes a glucosyl residue decorated with a xylosyl group, and "L" corresponds to a glucosyl residue decorated with a xylosyl group that is further connected to a galactosyl group. In previous literature, the "Q" has been described as "X," but we changed this in our model to avoid collision with the identifiers in arabinoxylan (Fig. 1 in Ravachol et al. [22]).

The latest studies of polysaccharide degradation by *R. cellulolyticum* are the degradation of arabinoxylan, as deciphered by Liu and colleagues (27, 42). This included the identification of an ABC importer dedicated to arabinoxylodextrins and its cognate intracellular degradation pathway. Arabinoxylan consists of a β-1,4 xylosyl unit backbone, branched with arabinosyl units (α-1,2 and/or α-1,3). The cellulosomes include xylanases and α-arabinofuranosidases that extracellularly break the polysaccharide into mono- and/or oligosaccharides (43, 44). However, the extent to which this polysaccharide is really degraded in the extracellular space during the growth of the bacterium remains elusive. Intracellular degradation is catalyzed by two α-L-arabinofuranosidases and two exo-xylanases, which target arabinose decorations and the xylose backbone, respectively (27). There are also esterases targeting the acetyl, feruloyl, and *p*-coumaroyl decorations (42), but these were not included in the model, as it is not determined how the acetyl groups decorate the xylose backbone. For the α-L-arabinofuranosidases and exo-xylanases, we included a reaction in the model for every metabolite following the experimental findings (Fig. 2). In the model, the letters X and A in the metabolite identifiers for the oligosaccharides indicate the type of decoration on a xylosyl residue in the backbone. "X" represents a xylosyl residue without any decoration, while "A" represents a xylosyl with an arabinosyl group. Some xylosyl residues have two arabinosyl groups, and the position on the xylosyl residue is given by a number (for example, XA2,3XX). In order to avoid gaps, we also included an α-L-arabinofuranosidase active on oligosaccharides XAX and XA2,3X with the respective gene associations. An ABC transporter was also included for every oligosaccharide that the solute binding protein can bind to.

## Manual curation of central carbon metabolism

We aimed to curate the model to perform well under multiple environmental conditions. This is especially important if the model is used in the context of microbial community modeling, where errors in individual models could lead to unrealistic interactions (45). The first curation step addressed central carbon metabolism. Since *R. cellulolyticum* is a strict anaerobe, the main energy gain occurs through fermentation pathways. Over the years, the species-specific enzymes and cofactors of *R. cellulolyticum* that are involved in glycolysis and fermentation have been studied (37, 46, 47). A literature survey was compiled into a data set of expected phenotypes (see supplemental material). Preliminary simulations using FBA and flux variability analysis (FVA) revealed that the production of the main fermentation products, acetate, ethanol, and lactate, was possible but not growth coupled. In genome-scale models of fermentative organisms, the way in which cofactors are balanced influences the predicted fermentation profile (48). Recent studies have shown that some of the kinases normally driven by ATP are instead driven by GTP or pyrophosphate (PPi) in *R. cellulolyticum* (11, 49). The same has been shown for *C. thermocellum* (50), leading to a predicted increase in ATP yield from glucose in both species (51). Furthermore, the fermentation pathways in species of Clostridia are known to be linked to the balancing of NADH, NADPH, and ferredoxin (52). We found that some reactions related to ferredoxin balancing were missing in the draft model, including NAD(P)-ferredoxin oxidoreductases, which we manually added.

## Manual curation of carbohydrate transporters

We then used the model to simulate gene deletions and compared the results with experimental data (11). This resulted in several false-positive growth predictions (Table 1), indicating that there were active pathways in the model that were not active or present in the organism, either due to regulatory effects or due to false annotations

in the model. By tracing the metabolic flux through the respective pathways, specific reaction targets were removed that would fix the false-positive results. We analyzed the evidence for these reactions and their respective proteins by comparing against high-confidence proteins in the UniProt database. This revealed several incorrectly annotated transport reactions in the model, including several phosphotransferase system (PTS) transporters. It has been shown that there are no active PTS transporters in *R. cellulolyticum* (53), but removing these reactions resulted in an incomplete pathway for galactose degradation, as the model no longer had a transport system for galactose. We found a candidate gene in UniProt (Ccel_1644) annotated as a galactose import ATP-binding protein. Although neither the role of this gene nor other transport mechanisms has been confirmed experimentally, the transporter was included in the model, since this mode of sugar transport is reported to be widely used by *R. cellulolyticum* (53). A UTP-glucose-1-phosphate uridylyltransferase was also included to complete the Leloir pathway for galactose metabolism (11). In addition to these false-positive predictions, the model incorrectly predicted that the gene encoding xylose isomerase (Ccel_3429) was essential for growth in xylose. Kampik et al. observed that this deletion was not lethal and hypothesized the presence of an alternative enzyme with xylose isomerase activity (11). Using UniProt, we discovered a second gene (Ccel_0500) that is predicted to encode another xylose isomerase. The curated model was able to correctly predict all the experimentally measured mutant phenotypes (Table 1).

## Calibration of maintenance energy

In the final curation step, the growth and non-growth-associated maintenance (GAM and NGAM) coefficients were calibrated using experimental data for chemostat cultivation at multiple dilution rates with cellobiose as the single carbon source (34) (Fig. 3a). The NGAM value has previously been determined to be 2.2 mmol/gDW/h for growth on cellobiose (17, 35). GAM was found using the same approach as for iFS431, where the sum of squared errors between the experimental and simulated growth rate at increasing uptake rates of cellobiose was minimized (see Materials and Methods). We also constrained the production of acetate, lactate, and ethanol to the experimentally determined values for each dilution rate. Due to solution degeneracy when simultaneously fitting GAM and NGAM, we opted to set NGAM to the reported value, whereas the GAM value was determined to be 30.22 mmol/gDW. Although this value is lower than the one used in the iFS431 model (40 mmol/gDW), it is close to a previously proposed value (35 mmol/gDW) estimated by Desvaux et al. (35).

Contrary to previous suggestions (11, 49), the use of GTP and PPi instead of ATP in some reactions in the model did not affect the maximum ATP yield (Table S1). To explore this in more detail, we constructed two central carbon metabolism models of glycolysis and fermentative pathways (core models) from the full iIB727 model: one using ATP in

**TABLE 1** Comparison between experimentally measured phenotypes of single-gene deletions grown on different sugars with model simulations (before and after manual curation) with indication of true-positive (TP), true-negative (TN), false-positive (FP), and false-negative (FN) predictions[a]

| Mutant | Enzyme | Sugar | Growth | Draft | Curated | Reference |
|--------|--------|-------|--------|-------|---------|-----------|
| MTL3221 | Hexokinase | Arabinose | Yes | TP | TP | (11) |
| MTL3238 | Galactokinase | Arabinose | Yes | TP | TP | (11) |
| MTL3429 | Xylose isomerase | Arabinose | Yes | TP | TP | (11) |
| MTL3431 | Xylulokinase | Arabinose | Yes | TP | TP | (11) |
| MTL2109 | Cellobiose phosphorylase | Cellobiose | No | FP | TN | (25) |
| MTL3221 | Hexokinase | Cellobiose | No | FP | TN | (11) |
| MTL3238 | Galactokinase | Galactose | No | FP | TN | (11) |
| MTL3221 | Hexokinase | Glucose | No | FP | TN | (11) |
| MTL3221 | Hexokinase | Mannose | No | FP | TN | (11) |
| MTL3429 | Xylose isomerase | Xylose | Yes | FN | TP | (11) |
| MTL3431 | Xylulokinase | Xylose | No | TN | TN | (11) |

[a]Experimental results for mutants are from Fosses et al. and Kampik et al. (11, 25).

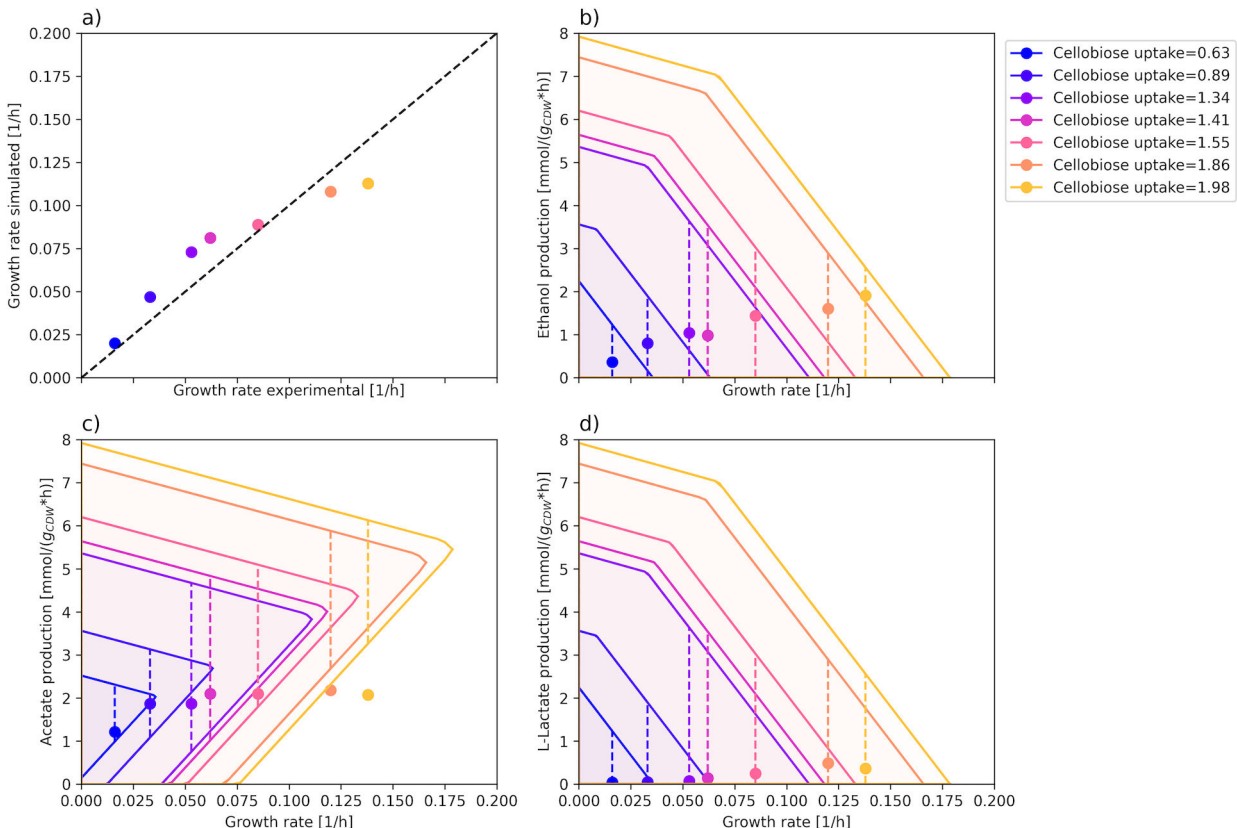

**FIG 3** (a) Experimental vs simulated growth rates with ilB727 model (with constraints on fermentation product production). The circles indicate the experimental and simulated growth rate at each uptake rate of cellobiose. (b–d) Production envelopes of the main fermentation products: acetate, ethanol, and L-lactate. The ranges are obtained with flux variability analysis in a carbon-limited scenario using the experimentally measured cellobiose uptake rate at various dilution rates. The circles indicate the measured production rates of each product at different growth rates, whereas the dotted lines show the flux variability at each growth rate. Experimental values are taken from Guedon et al. (34).

the hexokinase (HEX), phosphofructokinase (PFK), and pyruvate kinase (PYK) reactions, and another one with the updated cofactor preferences (Fig. S4). The ATP production rate was then maximized using flux balance analysis, revealing that the PPi and GTP used in the early steps of glycolysis (GTP in HEX and PPi in PFK) were regenerated further downstream at the expense of ATP. The first example of this is the PYK reaction, where GTP is being produced instead of ATP. Notably, a recent study confirms that PYK in *R. cellulolyticum* can use both GDP and ADP as cofactors (54). It is possible that PPi could also be rebalanced through PYK, but this has not been tested. According to our prediction, PPi was regenerated through pyruvate-phosphate dikinase (PPDK) operating in the direction from pyruvate to phosphoenolpyruvate, again at the expense of ATP. More investigation of the function of PPDK is needed, however, as it has not been studied for *R. cellulolyticum*, and it typically catalyzes the reaction in the opposite direction in heterotrophic bacteria (55).

## Fermentation profile

Using the curated model, we analyzed the profiles of the main fermentation products based on experimental cellobiose uptake rates from Guedon et al. (34) (Fig. 3b through d). In contrast to the draft model (Fig. S2b), the updated model predicted growth-coupled acetate production as the main fermentation product, which is in line with previous studies (34, 35). On the other hand, ethanol and lactate are not growth-coupled in the model. The predicted preference of the model for producing acetate over ethanol or lactate is likely caused by the increased ATP gain from acetate. The chemostat

experiments and analysis of enzyme activity by Guedon et al. (34) support this explanation, suggesting that acetate is favored when the energy source, cellobiose, is scarce. A similar analysis of iFS431 shows a similar pattern in growth coupling of acetate production (Fig. S3b through d).

The production of other fermentation products reflects the need to rebalance the NADH generated in glycolysis. Assays of intracellular metabolite concentrations and enzyme activity by Guedon et al. (34) suggest that, at low cellobiose uptake rates, redox balancing is primarily carried out by NADH-ferredoxin reductase, acetaldehyde dehydrogenase (ACALD), and alcohol dehydrogenase (ADH) (34). At higher uptake rates, the primary contributors shift to ACALD, ADH, and L-lactate dehydrogenase (LDH). Guedon et al. argue that this is due to oversaturation of pyruvate-ferredoxin oxidoreductase (PFO), leading to the build-up of intracellular pyruvate. Based on the enzyme assays, LDH is only active at high intracellular concentrations of pyruvate, since it has a high apparent Km in comparison to PFO. The shift in fermentation product production is visible from the experimental data in Fig. 3.

We can observe that the measured acetate production at the highest growth rate falls outside the computed production envelope (Fig. 3c). This discrepancy arises from the ATP maintenance coefficient, which was fitted to approximate the theoretical and experimental growth rates across all conditions. Although this improves the general applicability of the model, it may mean that the energy requirements at higher dilution rates are overestimated. In the iFS431 model, GAM was specifically fit to the lower growth rates, creating an even larger discrepancy from experimental data at higher growth rates (Fig. S3).

## Simulation of growth on cellulose and hemicellulose substrates

To assess the performance of our model under different conditions, we first replicated a batch culture experiment on cellulose, using dynamic FBA (dFBA) as originally performed by Salimi et al. (17), with experimental data from Desvaux et al. (56). In addition to the genome-scale model, the dFBA simulation requires initial conditions and kinetic equations for substrate uptake, which were defined similarly to Salimi et al. (17). The kinetic parameters for substrate uptake were optimized to fit the experimental data (see Materials and Methods), and the final values are presented in Table S2 along with other parameters for the simulation. Additionally, flux ratio constraints (see Materials and Methods) were required to reproduce the secretion of the non-growth-coupled fermentation products.

The results show that iIB727 is correctly reproducing the uptake, growth, and secretion profiles (Fig. 4). However, similarly to Salimi et al., we were unable to simulate the early termination of growth prior to the depletion of cellulose (Fig. 4a). This is expected because the mechanisms behind the growth termination are not included in the models. This effect is likely caused by the accumulation of toxic metabolites during growth on high concentrations of cellulose, which is a known phenomenon for *R. cellulolyticum* that was also pointed out by Salimi et al. (9, 17). The experimental results show that product secretion continues, at a lower rate, after growth arrest (Fig. 4b). This is not replicated by the models, since growth (and all metabolic activity) terminates upon substrate depletion. To replicate this effect, we could expand the model with inhibitory terms, but this is outside the scope of this work. We do observe a slightly early arrest of growth before cellulose is fully consumed. This happens when the biomass concentration is high, and the remaining substrate is not sufficient to fulfill the NGAM requirements for the next integration step.

Overall, the predicted concentration profiles qualitatively fit the experimental results for the period of exponential growth (solid lines in Fig. 4). Additionally, we can observe a low accumulation of glucose and cellobiose, which is also correctly predicted during the exponential phase (Fig. 4c). This likely happens due to the partial binding of the cells to the cellulose, which promotes the immediate use of released sugars and prevents their accumulation in the environment (57).

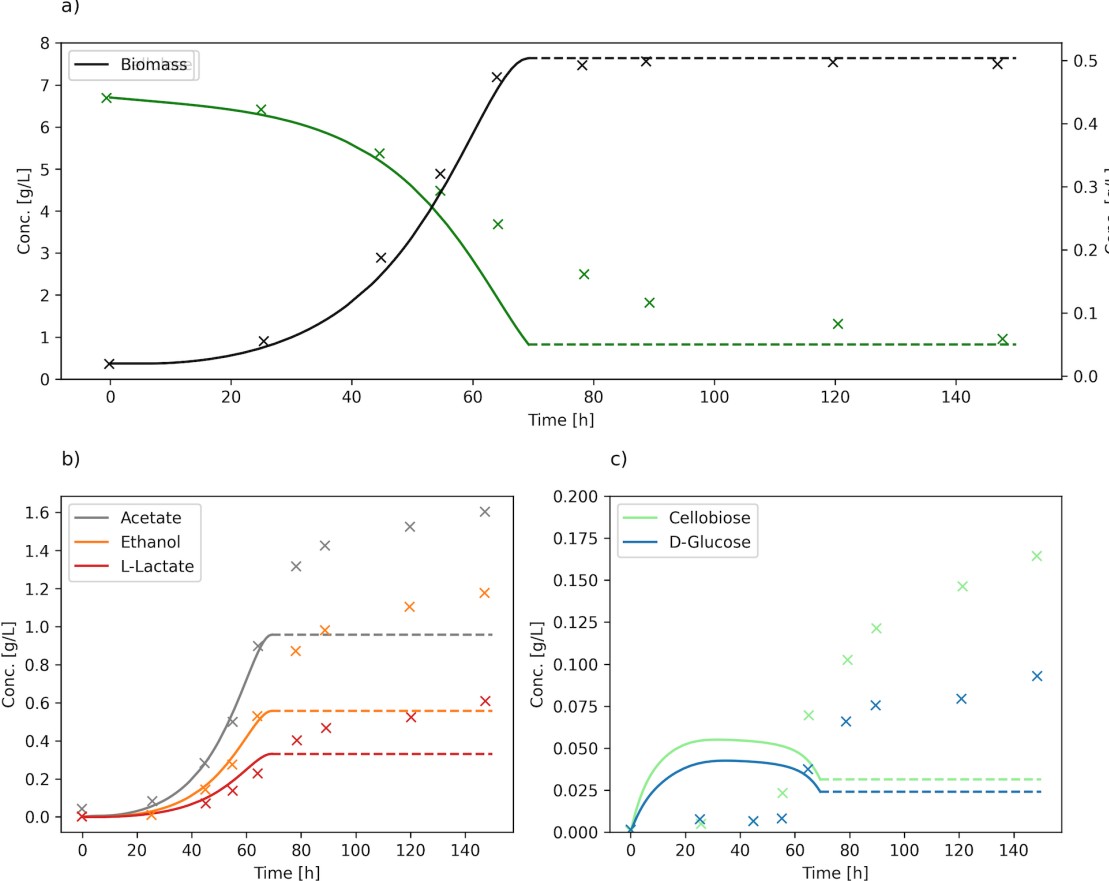

**FIG 4** Simulation of batch culture based on experiment by Desvaux et al. (56). (a) Cellulose and biomass concentration, (b) acetate, ethanol, and L-lactate concentration, and (c) cellobiose and D-glucose concentration in the medium. Dotted lines show the final concentrations after the simulation has terminated.

One of the main contributions from this work is the incorporation of pathways for cytosolic degradation of oligosaccharides from various mixtures of polysaccharides. To illustrate how this could be used in studies of polysaccharide degradation, we simulated growth on wheat straw, which is also a natural source of arabinoxylan. Based on the experimentally determined monosaccharide composition from Patyshakuliyeva et al. (58), we estimated the original polysaccharide composition. A previous study showed that *R. cellulolyticum* is able to simultaneously use various simple sugars, although with a preference for glucose (11). For the degradation of polysaccharides, however, the complexity increases since the structure of the substrate affects the accessibility of the degrading enzymes. A review on ruminal microorganisms uncovered that, in its pure form, hemicellulose is often amorphous and degrades faster than both hemicellulose integrated in the plant cell wall and cellulose (59). Therefore, due to a lack of suitable data on the kinetics of polysaccharide degradation, we assumed an evenly distributed degradation of polysaccharides with respect to each oligosaccharide (normalized by total carbon content).

Figure 5 shows the change in concentrations of polysaccharides, oligosaccharides, biomass, and fermentation products over an interval of 36 h. We can observe the simultaneous degradation of the polysaccharides and the accumulation of intermediate oligosaccharides of various sizes, followed by their depletion and secretion of the main fermentation products. Due to a lack of experimental data, we performed random sampling of the kinetic parameters using a normal distribution centered around the estimates from the cellulose batch culture simulation (see Materials and Methods). The time-course plots show the average concentrations and standard deviations across 100

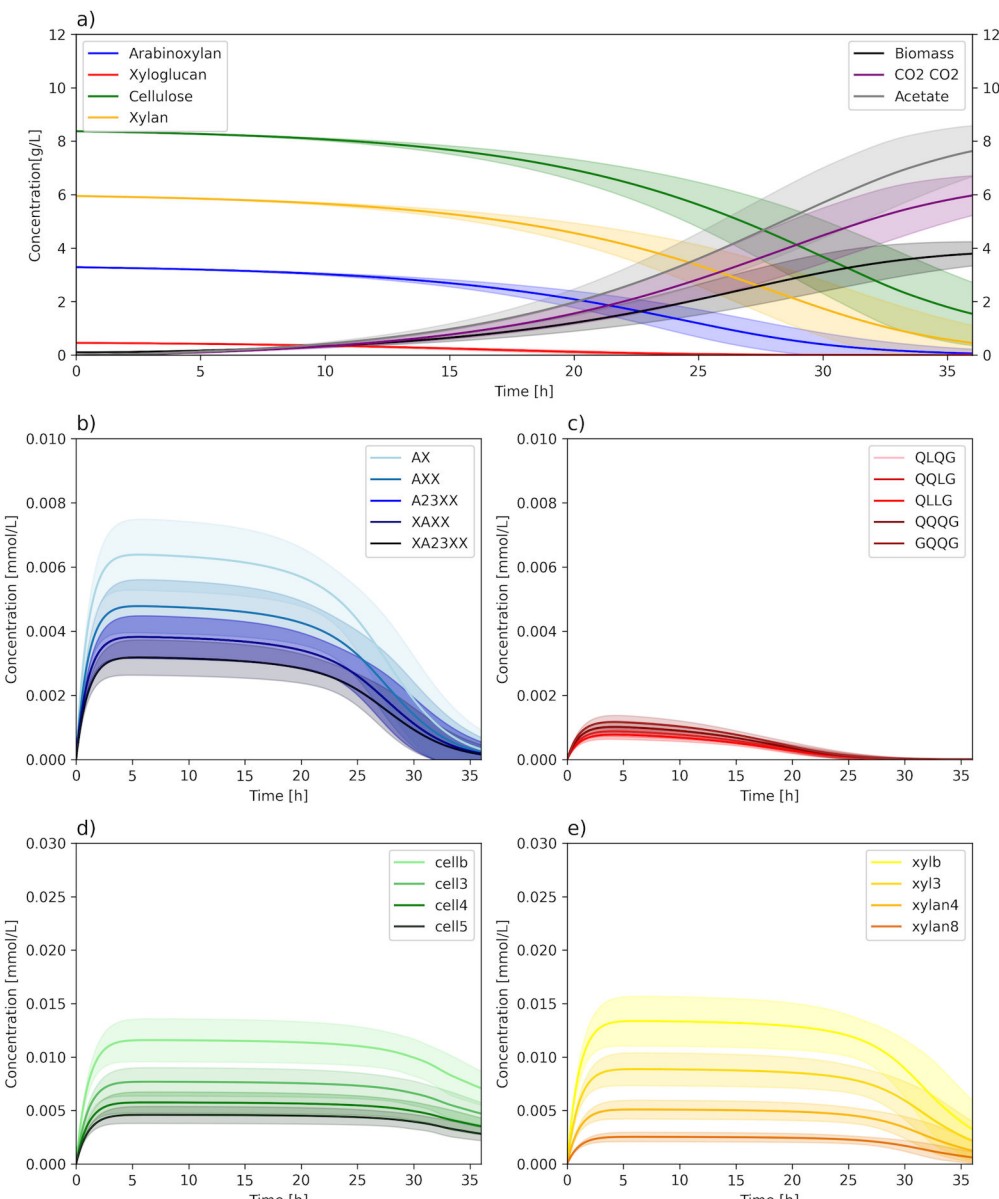

**FIG 5** Degradation of a lignocellulosic material from wheat straw. (a) Concentration of polysaccharides during a simulation of a batch experiment. The left *y*-axis shows polysaccharide concentration, while the right *y*-axis shows the biomass, acetate, and $CO_2$ concentration (g/L). (b–e) Concentration of multiple intermediate oligosaccharides generated from polysaccharide degradation and gradually consumed. The colors correspond to the original polysaccharide in panel a, sorted from smaller (lighter) to larger (darker) oligosaccharides. The line and error bands show, respectively, the mean and standard deviation across 100 simulations with random sampling of kinetic parameters.

simulations. We can observe that, despite some variability, the predicted qualitative behavior is quite consistent and robust to uncertainty in the parameters. This example shows how iIB727 can be used to predict the dynamics of complex polysaccharide degradation, especially as more data become available for parameter calibration.

## DISCUSSION

In this work, we reconstructed a genome-scale metabolic model for *R. cellulolyticum* strain H10 by combining automated reconstruction with manual curation based on experimental data. This model reflects an updated knowledge base aggregating two

decades of study of the metabolism of this bacterium. It incorporates a detailed description of the extracellular and intracellular pathways involved in the degradation and transport of lignocellulosic biomass components, including cellulose, xyloglucan, and arabinoxylan.

The initial draft model showed low production of the expected fermentation products. In anaerobic bacteria, cofactor balancing has a major effect on the final product. A common problem in genome-scale metabolic modeling is that cofactor usage cannot be predicted by homology and, in some cases, it is affected by the change of a single amino acid (60). There are available methods for predicting cofactor specificity that take into account the protein structure and amino acid residues close to the region of interest (61–63). In this case, we took advantage of experimentally determined cofactor dependencies previously reported (9, 11, 49). This alone greatly improved the prediction of fermentation profiles. On the other hand, the usage of GTP and PPi as cofactors in glycolysis did not have an impact on ATP yield. Our simulation also pointed to potential reactions that rebalance these cofactors at the expense of ATP, including PYK and PPDK. The predicted role of PYK fits recent studies, but the role of PPDK is yet to be confirmed for *R. cellulolyticum*. One recent study shows that PPDK also plays a key role in *C. thermocellum*, facilitating the conversion of phosphoenolpyruvate (PEP) to pyruvate, as this bacterium lacks genes for PYK (30). Interestingly, this organism also shows a preference for PPi and GTP in some of its glycolytic steps and is PPi-driven. This points to the role of PPi as an important energy currency, similar to what is usually observed for ATP. *R. cellulolyticum*, on the other hand, is believed to be mainly GTP-driven, since several enzymes in central carbon metabolism use GDP/GTP and show sensitivity to their concentration (54).

It has been previously shown that models generated with CarveMe have lower sensitivity in predicting gene essentiality in comparison to manually curated models, which is not unexpected for a fully automated method (31, 64). Despite the small size of the gene deletion data set used for validation, our results showed a similar pattern with several false-positive growth predictions (or, in other words, false-negative gene essentiality). This mainly resulted from the automated incorporation of alternative reactions for early steps of sugar catabolism and transport, especially the incorrect inclusion of PTS transporters. Transporter annotation is still one of the main aspects to address for improving automated reconstruction tools (65). Additionally, the model contained several reactions with more gene associations than reported in the literature. Tracing back the genes to the original model in the BiGG database revealed, in some cases, that the original annotation was also incorrect. The use of manually curated models as a basis for annotation is therefore not a guarantee for accurate and consistent automatic reconstructions.

In this work, we opted to model the degradation of the undigested polysaccharide as an equally distributed composition of oligomers. However, this degradation depends on where the hydrolyzing enzymes attach to the polysaccharide chain. This is a stochastic process that results in the formation of a pool of oligosaccharides of different sizes. Previous studies have proposed stochastic models for polysaccharide degradation (66, 67), which could be combined with genome-scale metabolic models. Such models could also include the localization of extracellular enzymes, as *R. cellulolyticum* produces both a cellulosome anchored to the cell wall and free extracellular cellulosomes (36). This would provide a more realistic description of polysaccharide degradation, although the computational cost might be high, and further development of stochastic models is needed, especially for branched polysaccharides. Furthermore, the mixed fermentation profile of *R. cellulolyticum* at higher growth rates indicates that protein constraints are likely at play. It is known that the availability and activity of enzymes are regulated by the concentration of the substrates, cofactors, and downstream products (11, 20, 26, 38, 49, 56). Tools like GECKO (68) can be applied to create a protein-constrained model based on our model. Methods have also recently been developed to create kinetic models from genome-scale metabolic models by network reduction (69). All in all, there are multiple

possibilities to expand the iIB727 model using more detailed modeling frameworks, provided that enough experimental data become available for estimation of kinetic parameters.

The main contribution of this study is the reconstruction of pathways involved in the degradation of lignocellulosic oligosaccharides. We have built pathways for the degradation of oligosaccharides from cellulose, xyloglucan, and arabinoxylan. Our model can be used to predict the metabolic phenotype of *R. cellulolyticum* while growing on lignocellulosic substrates with variable composition. Thompson et al. used the pathway for cellulose degradation in *C. thermocellum* to simulate changes in fermentation profiles on different cellodextrins (29). However, this bacterium is not able to grow on hemicellulose (12). Therefore, our model provides a valuable template for the reconstruction of other lignocellulose-degrading bacteria obtained from enrichment cultures. Such studies could be pivotal for understanding the role of *R. cellulolyticum* and other microbes in the optimization of the carboxylate platform. In a recent work (C. Schaefer et al., unpublished data), we reconstructed genome-scale models for organisms present in lignocellulose enrichment cultures and performed community simulations to reveal the underlying metabolic cross-feeding interactions between lignocellulose degraders and primary and secondary fermenters. Such simulations can contribute to the design of synthetic communities for the production of industrially relevant compounds, paving the way for a stronger bioeconomy based on renewable substrates.

## MATERIALS AND METHODS

### Reconstruction of draft model

The *R. cellulolyticum* strain H10 draft model was produced using CarveMe (version 1.5.1) (31) with the NCBI RefSeq genome assembly GCF_000022065.1. We used the universal model for gram-positive bacteria and gap-filling using 11 different growth media reported in previous studies (11, 22, 23, 26, 34–36). The media included were a mix of minimal (MM), defined (DM), and basal (BM) media, all of which have been reported to support the growth of *R. cellulolyticum*. MM is the simplest, containing inorganic elements in addition to riboflavin and nicotinate. DM contains additional vitamins. BM includes yeast extract and is therefore considered to be complex (36). As a substitute for yeast extract, we used the defined lysogenic broth medium from the media database in CarveMe, which includes all amino acids, vitamins, and nucleotides. The 11 variations of the media differ only by their main carbon source. As the pathways for oligosaccharide degradation were not present in the universal model, we substituted the polysaccharides by their constituent sugars.

### Manual curation steps

The multiple steps for manual curation consisted primarily of reconstructing the lignocellulose degradation pathways, modifying cofactor specificities according to literature, fixing incorrect gene essentiality predictions, and calibrating GAM/NGAM parameters. For reproducibility, all data extracted from the literature were compiled into spreadsheets, and all the model modification steps (adding and/or removing of genes, reactions, and metabolites) are encoded in Jupyter notebooks (available as supplemental material). At each curation step, we repeatedly ran a series of tests (using FBA and FVA) to confirm the ability of the organism to grow on the reported growth media and secrete the expected fermentation products, to search for blocked reactions, and to confirm the absence of energy-generating cycles that could be inadvertently introduced. Energy-generating cycles were checked by introducing and maximizing the energy dissipating reactions defined by Fritzmeier et al. (70) using parsimonious FBA (71). This minimizes the number of active reactions, simplifying the process of finding target reactions for curation. For simulations and reconstruction steps, both CobraPy (version 0.25.0) and reframed (version 1.2.1) were used.

For the final model, the Python package CobraMod (version 1.3.0) (72) was applied to add metabolite and reaction annotations based on the MetaNetX cross-reference database (73). UniProt records were used to add cross-references between gene/protein identifiers in UniProt, RefSeq, and KEGG. Some metabolite charges and chemical compositions were missing, and these were added manually based on entries in the BiGG database (74). The final version of the model was also tested using the MEMOTE suite (version 0.17.0) (33). Before running MEMOTE for the iFS431 model, we included the genes based on the supplementary material provided in Salimi et al. (17).

We calculated GAM through a constrained minimization (bounded between 10 and 50 mmol/gDW/h) of the sum squared error of the differences between experimental and simulated growth rates. The experimental data, taken from chemostat experiments by Guedon et al. (34), included growth rates, uptake rate of the limiting substrate (cellobiose), and secretion rates of the main fermentation products. The model was constrained with the rates of the substrate uptake and the fermentation product secretion, and growth rate was maximized.

To analyze the effects of cofactor preference, we created two versions of the central carbon metabolism model extracted from the full iIB727 model. These were named "Core models," and one uses ATP/ADP for the steps of glycolysis, while the other shows a nucleotide usage that reflects recent studies of *R. cellulolyticum*. This was done by copying elements directly from the curated iIB727 model and from the universal model in CarveMe. The function make_minimal_model (available in supplemental material) reproduces these steps in detail, and the complete map is presented in Fig. S4.

## Batch growth on cellulose

The modeling framework for dFBA simulation was based on the dFBAlab implementation in CobraPy (75, 76). We used the same mathematical formulation as given in Salimi et al. (17) for cellulose degradation, and glucose and cellobiose uptake rate. In their work, 1 mol of cellulose is broken down to 0.35 mol of cellobiose and 0.3 mol of glucose, which is defined separately from the genome-scale metabolic model. The experimental data used for validation were taken from Fig. 2 in Desvaux et al. (56) (using PlotDigitizer [67, 77]). Some changes were made to the method described by Salimi et al. to more accurately reproduce the experimental data. This included implementing flux ratio constraints for the main fermentation products (acetate, ethanol, and L-lactate) based on the experimental measurements. A flux ratio constraint was added between each fermentation product and the biomass reaction using the FBrAtio formulation (78). The maximization of growth rate, minimization of substrate uptake, and secretion of fermentation products were handled through the "lexicographic constraints" approach in the dFBAlab implementation, in that specific order. We fitted the kinetic parameters for uptake of cellobiose and glucose and cellulose degradation using differential evolution (SciPy 1.7.1) (79). The objective function minimized the distance (mean squared error) to the measured biomass and cellulose concentrations. The hyperparameter settings were as follows: mutation rate 0.5–1.0, recombination rate 0.7, population size 10, and maximum 50 iterations.

## Batch growth on multiple polysaccharides

As in the previous case study, the dFBA simulation was performed with dFBAlab. Since each polysaccharide can be degraded into a range of different oligosaccharides, the composition was adjusted so that each oligosaccharide contributes with an equal number of carbon atoms. The hydrolysis rates were scaled in terms of "glucose equivalents," consistent with the standard reporting of such rates in bacteria (17, 35, 80, 81) (the glucose equivalents for each polysaccharide are presented in Table S3). Thus, the hydrolysis reaction for a polysaccharide *i* that can be degraded into *n* oligosaccharides is given by

$$\text{poly}_i \rightarrow \frac{1}{n}\left(\frac{1}{\text{glc}_{\text{eq},1}}\text{oligo}_1 + \frac{1}{\text{glc}_{\text{eq},2}}\text{oligo}_2 + \ldots + \frac{1}{\text{glc}_{\text{eq},n}}\text{oligo}_n\right)$$

where $\text{glc}_{\text{eq}}$ is the number of glucose equivalents for each oligosaccharide. The respective hydrolysis rates were assumed to follow Michaelis-Menten kinetics:

$$v_{\text{poly},i} = \frac{v_{\text{max},i}\,C_{\text{poly},i}}{K_{m,i} + C_{\text{poly},i}}.$$

The specific production rate of an oligosaccharide $j$ was described by:

$$v_{\text{oligo},j}^{\text{out}} = \frac{v_{\text{poly},i}}{\text{glc}_{\text{eq},j}n_{\text{oligo},i}}, \forall j \in O_i$$

where $O_i$ is the set of oligosaccharides produced from polysaccharide $i$. The maximum uptake rate of an oligosaccharide $j$ was also assumed to follow Michaelis-Menten kinetics:

$$v_{\text{oligo},j}^{\text{in}} = \frac{v_{\text{max},j}\,C_{\text{oligo},j}}{K_{m,j} + C_{\text{oligo},j}}.$$

The default values used were $v_{\text{max},i} = 1$ (mmol/gDW/h), $K_{m,i} = 5$ (mmol/L) for every polysaccharide and $v_{\text{max},j} = 10$ (mmol/gDW/h), $K_{m,j} = 1$ (mmol/L) for every oligosaccharide. To analyze the sensitivity of the simulation to changes in the parameters, we ran 100 simulations with random perturbations ($p$) to the default parameter values ($p_0$) following a log-normal distribution:

$$p = p_0\,X,\ \ln(X) \sim N(0,1).$$

## ACKNOWLEDGMENTS

The authors would like to thank the suggestions from Radhakrishnan Mahadevan on the manuscript.

This work was supported by project Cell4Chem, funded under the 3rd ERA CoBioTech Joint Transnational Call, with additional support from the Centre of Digital Life Norway (DLN), funded by the Research Council of Norway.

## AUTHOR AFFILIATIONS

[1]Department of Biotechnology and Food Science, Norwegian University of Science and Technology (NTNU), Trondheim, Norway

[2]Faculty of Biosciences, Norwegian University of Life Sciences (NMBU), As, Akershus, Norway

[3]Oslo Centre for Biostatistics and Epidemiology (OCBE), Oslo University Hospital, Oslo, Oslo, Norway

[4]Laboratoire de Chimie Bactérienne (LCB), Marseille, Provence-Alpes-Côte d'Azur, France

## AUTHOR ORCIDs

Henri-Pierre Fierobe https://orcid.org/0000-0003-0468-7180
Daniel Machado http://orcid.org/0000-0002-2063-5383

## FUNDING

| Funder | Grant(s) | Author(s) |
| --- | --- | --- |
| Horizon 2020 Framework Programme | 722361 | Idun Burgos |

| Funder | Grant(s) | Author(s) |
|---|---|---|
| | | Stéphanie Perret |
| | | Henri-Pierre Fierobe |
| | | Daniel Machado |

## AUTHOR CONTRIBUTIONS

Idun Burgos, Data curation, Formal analysis, Investigation, Methodology, Software, Validation, Visualization, Writing – original draft | Ove Øyås, Investigation, Methodology, Supervision, Writing – original draft | Stéphanie Perret, Validation, Writing – original draft | Henri-Pierre Fierobe, Validation, Writing – original draft | Daniel Machado, Conceptualization, Funding acquisition, Methodology, Project administration, Supervision, Writing – original draft

## DATA AVAILABILITY

The curated version of the iIB727 model is publicly available at BioModels.net (MODEL2503030001). All supplemental data and code are publicly available at https://github.com/IdunBurgos/Ruminiclostridium-cellullolyticum-model-final.

## ADDITIONAL FILES

The following material is available online.

### Supplemental Material

**Supplemental material (mSystems00960-25-s0001.pdf).** Supplemental figures and tables.

### Open Peer Review

**PEER REVIEW HISTORY (review-history.pdf).** An accounting of the reviewer comments and feedback.

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
