## [Reviewer comments · mSystems]

Genome-scale metabolic modeling of *Ruminiclostridium cellulolyticum*: a microbial cell factory for valorization of lignocellulosic biomass

Idun Burgos, Ove Øyas, Stéphanie Perret, Henri-Pierre Fierobe, and Daniel Machado

Corresponding Author(s): Daniel Machado, Norges teknisk-naturvitenskapelige universitet

Review Timeline:

Submission Date:	June 27, 2025
Editorial Decision:	August 1, 2025
Revision Received:	August 19, 2025
Accepted:	September 10, 2025

Editor: Hans Bernstein

Reviewer(s): The reviewers have opted to remain anonymous.

Transaction Report:

DOI: <https://doi.org/10.1128/msystems.00960-25>

Re: mSystems00960-25 (**Genome-scale metabolic modeling of *Ruminiclostridium cellulolyticum*: a microbial cell factory for valorization of lignocellulosic biomass**)

Dear Prof. Daniel Machado:

Thank you for the privilege of reviewing your work. Below you will find my comments, instructions from the mSystems editorial office, and the reviewer comments. Please note that both reviewers were satisfied with your revisions but that Reviewer #1 asked to see your responses in one more round of minor revisions. I agree that this will lead to an improved manuscript.

Revision Guidelines

Sincerely,
Hans Bernstein
Editor
mSystems

Reviewer #1 (Comments for the Author):

The authors addressed most of my previous comments. I still have a few additional points that need clarification.

1. Incorporating GAM into the biomass equation via the ATP coefficient makes sense. However, there may be additional ATP requirements for the synthesis of biomass precursors. This ATP demand is proportional to biomass formation - so it is also

"growth-associated." Is this not considered part of GAM then? If yes, the ATP coefficient in the biomass equation may underrepresent GAM. If not, what is the definition of GAM in this context?

2. "The optimization of growth rate, substrate uptake, and secretion of fermentation products was handled through the 'lexicographic constraints' approach in the dFBA implementation in that specific order." - I have several comments and questions:

- a. The current description is unclear. Instead of "optimization," it would be more explicit to state "maximization of growth rate followed by minimization of substrate uptake and product secretion," as described in the response document.
- b. Biomass maximization typically occurs under constrained substrate uptake. How is it possible to minimize substrate uptake after biomass maximization?
- c. In most FBA simulations, minimizing product secretion results in no product formation. Why is that not the case here?
- d. When multiple products are involved, lexicographic optimization should specify the order of product minimization. Please provide the order used and the rationale behind it.

3. Questions regarding Figs. 3, S2, and S3: Panels (b) and (c) show variations in predicted growth rates and product secretion. How did the authors determine the specific growth rate values indicated by the dashed lines? Similarly, how were the values in panel (a) in each figure determined?

4. Measured growth rates are missing in Fig. S2(a).

Reviewer #2 (Comments for the Author):

Thank you to the authors for thoughtfully considering and addressing the comments on the original submission; I find the revisions to add substantially to the clarity, substance, and potential application of the manuscript. While I have no further comments for content improvement, it seems that the revision process introduced a number of additional grammatical errors; I would recommend a more careful proofreading before any next steps. Additionally, the added ending to the last paragraph of the Discussion ("In a recent work (Schaefer et al., in preparation), we reconstructed..."), feels like a promotion of the authors' own work, whereas the Discussion should conclude with a broader statement of application to the general scientific community/field. I recommend improving transitions (e.g., "For example, in a recent work..."), adding examples of other hypotheses that could be tested with this model/platform, and reorganizing sentences so that the last paragraph of the Discussion ends on a broader note.

Response to reviewers

Once again, we thank the reviewers for the feedback and positive contributions to our manuscript.

Reviewer #1

1. Incorporating GAM into the biomass equation via the ATP coefficient makes sense. However, there may be additional ATP requirements for the synthesis of biomass precursors. This ATP demand is proportional to biomass formation - so it is also "growth-associated." Is this not considered part of GAM then? If yes, the ATP coefficient in the biomass equation may underrepresent GAM. If not, what is the definition of GAM in this context?

The GAM and NGAM parameters represent the energy that is spent both in metabolic and other biological processes that are not explicitly accounted for in the model. Indeed, this includes the assembly of biomass precursors (proteins, lipids, RNA molecules), plus all the transcriptional and translational machinery and any mechanisms associated with cell division. Given such complexity, the overall values for these parameters are estimated empirically, usually following the protocol proposed by Thiele and Palsson (Nature Protocols, 2010). Using chemostat data at multiple dilution rates, where both the substrate uptake and growth rates are known, one can calibrate the parameters such that the model fits the experimental data.

2. "The optimization of growth rate, substrate uptake, and secretion of fermentation products was handled through the 'lexicographic constraints' approach in the dFBA lab implementation in that specific order." - I have several comments and questions:

a. The current description is unclear. Instead of "optimization," it would be more explicit to state "maximization of growth rate followed by minimization of substrate uptake and product secretion," as described in the response document.

We modified the text as suggested.

b. Biomass maximization typically occurs under constrained substrate uptake. How is it possible to minimize substrate uptake after biomass maximization?

Unfortunately, the available software implementation of dFBA lab only exports simulation results for the variables that are included in, what its authors call, the objective function with "lexicographic order". In fact, as the reviewer points out correctly, since biomass optimization occurs under constrained substrate uptake, minimizing substrate uptake has no real effect on the final result. It is just a workaround we had to use due to this limitation in the software.

c. In most FBA simulations, minimizing product secretion results in no product formation. Why is that not the case here?

In this case, since the growth is anaerobic, some amount of fermentation (mainly acetate) is necessary. However, as explained previously, we only added product formation to the lexicographic constraints

due to the limitation mentioned above. Like explained in the main text, we added flux-ratio constraints to couple growth and secretion of the main fermentation products (acetate, ethanol, lactate). Therefore, the secretion rates are not really influenced by their presence in the objective function.

d. When multiple products are involved, lexicographic optimization should specify the order of product minimization. Please provide the order used and the rationale behind it.

To improve clarity, we have specified the order. But, as mentioned above, it does not really influence the final result.

3. Questions regarding Figs. 3, S2, and S3: Panels (b) and (c) show variations in predicted growth rates and product secretion. How did the authors determine the specific growth rate values indicated by the dashed lines? Similarly, how were the values in panel (a) in each figure determined?

The specific growth and production rates are reported in the paper by Guedon et al. (1999). This was only mentioned in the main text, but we have now made it clearer in the Figure text as well.

4. Measured growth rates are missing in Fig. S2(a).

Thank you, this is now fixed.

Reviewer #2

Thank you to the authors for thoughtfully considering and addressing the comments on the original submission; I find the revisions to add substantially to the clarity, substance, and potential application of the manuscript. While I have no further comments for content improvement, it seems that the revision process introduced a number of additional grammatical errors; I would recommend a more careful proofreading before any next steps. Additionally, the added ending to the last paragraph of the Discussion ("In a recent work (Schaefer et al., in preparation), we reconstructed..."), feels like a promotion of the authors' own work, whereas the Discussion should conclude with a broader statement of application to the general scientific community/field. I recommend improving transitions (e.g., "For example, in a recent work..."), adding examples of other hypotheses that could be tested with this model/platform, and reorganizing sentences so that the last paragraph of the Discussion ends on a broader note.

We followed this suggestion and rephrased the final paragraph as follows:

In a recent work (Schaefer et al, in preparation), we reconstructed genome-scale models for organisms present in lignocellulose enrichment cultures and performed community simulations to reveal the underlying metabolic cross-feeding interactions between lignocellulose degraders and primary and secondary fermenters. Such simulations can contribute to the design of synthetic communities for the production of industrially relevant compounds, paving the way for a stronger bioeconomy based on renewable substrates.

Re: mSystems00960-25R1 (**Genome-scale metabolic modeling of *Ruminiclostridium cellulolyticum*: a microbial cell factory for valorization of lignocellulosic biomass**)

Dear Prof. Daniel Machado:

Thank you for addressing the reviewer's concerns through multiple rounds of edits.

Your manuscript has been accepted, and I am forwarding it to the ASM production staff for publication. Your paper will first be checked to make sure all elements meet the technical requirements. ASM staff will contact you if anything needs to be revised before copyediting and production can begin. Otherwise, you will be notified when your proofs are ready to be viewed.

Sincerely,
Hans Bernstein
Editor
mSystems

Reviewer #1 (Comments for the Author):

The authors addressed all the issues raised and I have no further comments.